# Living a Healthy Life in Australia: Exploring Influences on Health for Refugees from Myanmar

**DOI:** 10.3390/ijerph17010121

**Published:** 2019-12-23

**Authors:** Carrie K. Wong, Carolynne White, Bwe Thay, Annie-Claude M. Lassemillante

**Affiliations:** 1School of Health Sciences, Swinburne University of Technology, Hawthorn 3122, Australiaalassemillante@swin.edu.au (A.-C.M.L.); 2Office of Pro-Vice Chancellor (Student Engagement), Swinburne University of Technology, Hawthorn 3122, Australia

**Keywords:** migrants, refugees, asylum seekers, adults, health promotion, primary health care, community-based participatory research, focus groups, social support, work, education, access to healthcare

## Abstract

Background: Humanitarian migrants from Myanmar represent a significant refugee group in Australia; however, knowledge of their health needs and priorities is limited. This study aims to explore the meaning and influencers of health from the perspectives of refugees from Myanmar. Method: Using a community-based participatory research (CBPR) design, a partnership was formed between the researchers, Myanmar community leaders and other service providers to inform study design. A total of 27 participants were recruited from a government-funded English language program. Data were collected using a short demographic survey and four focus groups, and were analysed using descriptive statistics and thematic analysis methods. Results: Key themes identified included: (1) health according to the perspectives of Australian settled refugees from Myanmar, (2) social connections and what it means to be part of community, (3) work as a key influence on health, and (4) education and its links with work and health. Conclusions: This study outlined the inter-relationships between health, social connections, work and education from the perspectives of refugees from Myanmar. It also outlined how people from Myanmar who are of a refugee background possess strengths that can be used to manage the various health challenges they face in their new environment.

## 1. Introduction

A significant number of refugees settle into new home countries every year. In 2018, Australia accepted 16,250 people from refugee backgrounds (refugee status as defined by the United Nations High Commissioner for Refugees) through the Humanitarian Program [1]. The top five countries of birth of humanitarian migrants in 2017–2018 were Iraq, Syria, Myanmar, Congo (DRC), and Afghanistan [1]. Many refugees arrive in their new home country after having experienced trauma [2,3,4,5], and often continue to face health and economic disadvantage in their new home country [6,7,8,9,10].

People from refugee backgrounds have poorer health outcomes than those from non-refugee backgrounds [6,7,8,9,11,12,13]. Lack of access to health services in their new home country [10,14,15] presents an additional challenge that is influenced by language barriers, lack of transport, and unfamiliar health service systems [2,16,17,18,19,20]. Several studies suggest that people from refugee backgrounds have low health literacy, which poses both a barrier and a deterrent to accessing health services [16,21,22,23,24,25,26,27]. Difficulties accessing health services is compounded by the difference in cultural beliefs among each refugee community group and how culture may influence their view of health [16,21,25,28].

One framework used to understand the health of refugees is the socio-ecological model (SEM), as it recognises that the health of individuals is influenced by complex interactions between social, cultural, and environmental factors [29,30,31,32,33]. To date, several studies have applied the SEM to understand the impacts of social and environmental influences on access to healthcare and health behaviour amongst migrant and refugee groups [30,31,32,33]. It is however key that such research efforts involve migrant communities to remain culturally grounded and reflect people’s lived experiences.

Knowledge of the health needs and priorities of refugees from Myanmar who reside in Australia is currently limited. Indeed, community members report health promotion programs may be ineffective due to lack of community engagement and consultation. In this paper, we describe how community-based participatory research (CBPR) methods were used to engage and build a relationship with the Myanmar refugee community as the foundation for future community-led health promotion efforts. The aim of this study was to investigate the meaning of and influences on health from the perspectives of refugees from Myanmar.

## 2. Methodology and Methods

Acknowledging trauma and the potential for participants to be re-traumatised through their engagement in research is a key consideration when conducting ethical research with refugee participants [34]. Due to historical events and political influences in Myanmar, it was likely that potential participants had experienced trauma, either in their home country or en route to Australia. Consequently, the study design adopted a trauma-informed approach, which focused on the community’s strengths and incorporated the principles of safety, trustworthiness, choice, collaboration and empowerment [35]. CBPR aligns with a trauma-informed approach and uses the community’s strengths to address an issue of importance and help effect change [36].

A CBPR design and qualitative methods were chosen to investigate experiences of health and its determinants among refugees from Myanmar. Although qualitative research methods aim to amplify the voice of participants [37], CBPR has been used effectively within migrant communities to develop effective interventions to address health inequalities [38]. The CBPR process involved establishing a partnership between the researchers and community members, before obtaining ethical approval to conduct the study from the Swinburne University Human Research Ethics Committee (#2017/19).

### 2.1. Partnership

The research team was approached by a community leader who recognised the need for a health promotion program for refugees from Myanmar. A research partnership between community leaders, the local primary health care service and Swinburne University of Technology was formalised and a steering committee was established to guide the study. The steering committee met every three months and included key stakeholders including the researchers (dietitian, occupational therapist), two clinicians from the local community health service (dietitian, doctor), a nurse from the campus health service, two student members of the refugee community (referred to as ‘community members’), the campus Adult Migration English Program (AMEP) convenor and a representative from student engagement and the Migrant Information Centre. Four members of the steering committee were from Myanmar.

The purpose of the steering committee was to: (i) gain an overview of the issues of concern to the community; (ii) develop the research questions, recruitment material and data collection tools; (iii) engage the relevant refugee community and (iv) interpret and disseminate the findings from the study. Minutes were taken at each steering committee to document discussions and decisions and provide an audit trail for the project. At the beginning of the project, a member of the steering committee who was from Myanmar and had refugee experience (BT) provided cultural awareness training. This gave the researchers a contextual overview of life in Myanmar, common refugee experiences and insights into relevant cultural beliefs and practices.

### 2.2. Setting and Recruitment 

Swinburne University of Technology (SUT) is a dual sector education provider that delivers English classes to refugees across three campuses in Melbourne, Australia. The AMEP at the Croydon campus was selected as the setting for this study because (i) 2.3% of the total population of this suburb is from Myanmar and (ii) more than 50% of the student cohort were humanitarian migrants from Myanmar. Purposive sampling was used to recruit students from intermediate (level 2) and higher level (level 3) AMEP classes, as these students had some English language proficiency. These students were also more likely to provide in-depth information as they had spent more time living in Australia and could reflect on their settlement. Inclusion criteria were (i) from Myanmar; (ii) enrolled in AMEP at SUT, Croydon campus and (iii) aged between 18 and 65 years. Students were excluded if (i) they were born in Australia or a country other than Myanmar, (ii) they were unable to give informed consent due to mental or cognitive impairment or (iii) they were aged under 18 years or over 65 years.

Prior to recruitment, rapport with the students was built over time via a person trusted by the community. The campus nurse introduced the researchers (A.L. or C.W.) to students. Over the next month, the researchers attended classes to build relationships and trust with students in a familiar and safe environment. During this time period, a local healthcare provider (EACH) delivered one health education session as part of their comprehensive refugee primary healthcare service model, which was developed to align the research with the AMEP curriculum. One week prior to data collection, the researchers (A.L. and C.W.) and the campus nurse attended level 2 and 3 AMEP classes to explain the research project, with assistance from one of the steering committee members as an interpreter. All students from Myanmar were invited to participate.

### 2.3. Data Collection

A short demographic survey and a focus group topic guide were co-designed by the steering committee. Information regarding refugee status was not collected, as all students enrolled in the AMEP were holders of a humanitarian migrant visa, and therefore their refugee status was assumed. On the morning of the focus group, students were collected from their class. The participant information and consent statement was read out and translated into the relevant community language by a qualified interpreter, who had been briefed on the purpose of the study and focus group topics. Three students declined to participate and returned to their classroom. After giving consent, participating students were given a $20 gift voucher to reimburse them for any expenses associated with their involvement in the study.

A total of 27 students gave verbal consent to participate, and completed a short written demographic questionnaire, independently or with assistance from an interpreter. Four focus groups, of 60 min duration, were conducted with between six to eight participants. The focus groups were conducted in English and moderated by the lead researcher (C.W.), according to a semi-structured topic guide. Students could respond in English or in their preferred language via an interpreter. The focus groups were audio-recorded using a digital audio recorder, and a note-taker (A.L.) recorded written observations of participants, seating arrangements and body language to provide context and help interpret the verbal information. After the focus groups, the moderator and note-taker offered the interpreters an opportunity to debrief. During this time, some interpreters provided contextual information, which gave the researchers a deeper understanding of the participants’ responses and the issues that arose during the focus groups. Once the interpreters had gone, the researchers wrote field notes separately, before discussing their observations and insights together.

### 2.4. Data Analysis

Quantitative data from the demographic questionnaire were analysed using descriptive statistics using Microsoft Excel. Audio-recordings from the focus groups were transcribed verbatim by two researchers (A.L. and C.W.). The focus group transcripts were analysed using thematic analysis techniques [37,39]. For each focus group, two researchers (A.L. and C.W.) listened to audio recordings and read the transcripts to become familiar with the data. Each researcher wrote fieldnotes documenting their first impressions and observations. Then, transcripts were coded inductively before constructing candidate themes through a process of collating and refining codes and mapping potential themes. The researchers analysed data from the first focus group together to develop candidate themes before analysing the subsequent focus groups separately. Once each focus group was coded and codes were collated into themes, two researchers (A.L. and C.W.) reviewed and compared the themes from each focus group to further develop the themes. The candidate themes were presented to the research steering committee members and revised in response to their feedback. The components of the SEM were used to develop the major themes and findings are presented across three levels of this model.

### 2.5. Rigour

Strategies to enhance trustworthiness were incorporated into the study design [40]. Minutes of steering committee meetings and emails between the researchers provided an audit trail throughout the project. Following each focus group, each researcher wrote detailed field notes and then compared them through peer debriefing. To further enhance credibility, member checking was conducted, whereby a summary of themes were presented and discussed at a steering committee meeting and at a lunch for student participants hosted by the research team. Students present at the member checking session may or may not have been involved in data collection due to the short-term nature of the English classes program and unpredictable class attendance. However, the confirmation of main themes by the independent student group enabled us to ensure the accuracy and validity of our research. All relevant information or feedback provided by steering committee members or participants was incorporated into the themes. Demographic information was collected and used to describe the participants to assist readers to transfer the findings to other settings.

## 3. Findings

A total of 27 participants were recruited, including 12 men and 15 women from the SUT AMEP classes. Of the participants, 23 were recruited from intermediate English level classes (level 2), whilst four were recruited from higher level English level classes (level 3). Participants spoke a variety of Myanmar languages including Zomi, Haka Chin, Burmese, and Karen. More than half (66.7%) of participants had lived in Australia for more than 2 years. Participant characteristics are presented in Table 1.

### 3.1. Health According to the Perspectives of Australian Settled Refugees from Myanmar

Participants identified good health as being relevant to every person regardless of where they lived, and related health to different domains such as physical, social and mental health. One participant described health holistically as “your mind, your spirit and your body”. Physical health related to feelings of wellness, such as energy levels and absence of pain, whilst mental health was associated with “not overthinking” and having “a peaceful heart”.

Participants discussed practical health behaviours that were conducive to the attainment of health including eating, physical activities and strategies for managing mental health. As one participant described, “health means to me eating regularly, sleeping regularly and also peaceful mindful heart not to overthink”. Eating behaviours, which included consuming fruits and vegetables, drinking less alcohol, and reducing fat and salt intake, were identified by participants as supporting their health.

The physical health of participants was influenced by their migration journey, and participants made comparisons in the differences in healthcare access before and after arrival in Australia. For instance, participants compared healthcare access in Myanmar and places of transit with the healthcare that was available in Australia. In Myanmar, accessing healthcare services was difficult, especially for participants who lived in villages, where the government provided infrastructure but not services, including staff and equipment. Limited finances were an additional barrier to seeking healthcare. One participant commented, “[We] cannot easily go to the doctors since we do not have enough money”.

Once participants arrived in Australia, they received free comprehensive health screenings and follow-up as part of the settlement process. Overall, participants appreciated the level of healthcare available to them in Australia, with one participant saying, “the care in Australia in health departments is really good”, and another stating, “we’ve got full assistance by [government funded healthcare scheme], and we’ve got full support for everything that we need”. The significance of accessing more comprehensive healthcare in Australia was highlighted by one participant who said, “the health system’s changing. When [participants] were in the camp, they didn’t have enough access to healthcare, so they got sick and they don’t have enough access to medications, and when they came here they prescribed with the right medication”.

However, participants experienced challenges understanding and using the Australian health system when compared with other countries they had lived in before. This was illustrated by one participant who described, “in our country you can buy medicine without prescription so that is different because [in Australia] they always investigate it first and never prescribe medication easily… so that is really hard to understand.”

Another participant raised concerns about the need to make a healthcare appointment in Australia, “sometimes when we get sick and then make an appointment and we can’t get the same date so wait another two or three days”. Managing multiple health appointments created additional challenges. For instance, one participant described, “Sometimes getting around is a problem for me for example a doctor refer me to a certain hospital or clinical service but I don’t know the location and sometimes I fail to attend for example once I need to check my stomach and get a gastroscopy but they give me a certain place and I do not know where to get there.” For newer members of the community, lack of familiarity with geographical locations, limited access to transport and language barriers resulted in missed healthcare appointments.

### 3.2. Social Connections: “It Is Good to Be Part of Community”

Participants perceived social connection as a key part of health within their culture, “we tend to live together in a group, very closely”. These social networks included family, community members from their ethnic group, church and the broader Australian community. Social connections therefore played an important role in achieving good social and mental health, as one participant explained, “I think it is good to be a part of community surely it is better than living alone or having no one to talk to. And it’s good if you can go to church every week with a set of people and if you participate so I think that it is good for your mental health and social”.

Upon arrival in Australia, people from Myanmar sought to live where other community members of the same ethnicity resided, and this was outlined by one participant who said, “most community members tend to move where their community members already there so it is very important for community.” Being able to connect with people from the same ethnicity and language group through activities such as going to church, celebrating personal and community events and playing sports was therefore viewed as important. After arriving in Australia, participants also connected with the broader Myanmar community, as illustrated by one participant, “only Chin is big enough, Chin community but we have other connections as well as the other ethnicities from Burma”. Connections between people from different ethnic groups were generally made through organised multi-ethnic Myanmar events, which afforded smaller ethnic groups the opportunities to connect with others.

Social connection was a key resource used by participants to overcome challenges, and to adapt to life in Australia. For instance, participants supported each other to engage in health-promoting behaviours and worked together to address challenges within their communities, “Others do [smoking, drink alcohol, chew betel nut], if it’s possible for us to come together even if it’s to reduce the amount of intake”. Social connection was also a source of emotional support, as one participant explained, “sometimes everyone like feel sad and have problems so if they think about themselves the problem gets worse so it’s good to share with friends and um if you get with others we can listen to them, support them and encourage them so that’s how they relieve their problems.”

Participants shared information with new arrivals about local resources such as where to get food from “both from supermarket, from the garden, from Burmese grocery shop” and opportunities for physical activity. Focus group participants reported engaging in a variety of activities such as riding a bike, going to the gym, group sports such as soccer and volleyball, gardening, and casual dancing. These activities were done in a social context, that is, with family members, friends or other members of the Myanmar community. However, their choice of physical activity was influenced by their gender and associated roles. As one participant explained, “[no time for exercise] because my husband is always at work so I am the one that has to look after the children”, whereas another participant added, “because I have a young kid, it’s hard for me to go out and exercise”. Therefore, while men engaged in social sports, such as playing soccer, which required planning, women engaged in spontaneous physical activities such as going to the local park to play volleyball and casual dancing at home.

Community members also shared information about navigating the health system with new arrivals. One participant explained, “I tell [new arrivals] the health system in Australia because in our country we just go to see a doctor, just pop in, [I] just tell them that we need to make an appointment first and see a doctor”. Community members who have been settled for longer periods in Australia therefore provided practical assistance to new arrivals such as taking them to medical appointments.

### 3.3. Work: A Key Influence on Health

The ability to work was a key influence on the participants’ health. The participants described a strong work ethic, as one man commented, “we are born to work and eat”. Another participant explained, “back in Myanmar, I work and walk a lot”. Working directly benefited the participants’ physical health, for instance earning money enabled participants to afford a wider variety of foods, and also provided incidental exercise when travelling to and from work or when performing physically demanding jobs. Many participants experienced barriers to paid employment since their arrival in Australia.

Participants described the frustration of being “willing to work, but we can’t get a job”. Whilst many participants had skills and work experience, participants felt that these were not recognised or relevant in Australia, “we don’t have any skills. Um, even if we [have] work experience, it’s not related in this country, we can’t really use them”. Some participants had received support from a job network provider to gain local experience, “we got a lot of support so even though they provide volunteering, um I have done volunteering for a week, I still can’t get a job…” The distance required to travel to available job opportunities was also reported to be a hindrance to employment, with one participant saying, “there may be some [employment] but it is too far and [Myanmar community members] can’t go”.

Most participants relied on financial support from Centrelink (government welfare support), which gave participants some means to support their health through buying food or going to the doctor. Despite this financial support, some participants reported financial hardship, which affected both their physical and mental health. To overcome this hardship, some participants had established a garden at their home so they could pick fresh vegetables to eat. In addition to ensuring a healthy diet, gardening and doing other household work had additional benefits by “mak[ing] yourself as busy as you can so you can sleep better.” One participant commented, “we are working unpaid job, full-time job at home… looking after children, and gardening.”

### 3.4. Education: Links with Work and Health

English proficiency was perceived as necessary to secure employment in Australia, and all participants were enrolled in English language classes. As one participant said, “To get a job, we need experience and speak English in certain case”. One participant explained this further, “I think that language could be a bigger barrier than qualifications because in some fields, such as working in a factory, you don’t necessarily need any qualifications…but you do need the language so if you speak English getting to that employment then you’ll be able to pick up on that trade.”

Participants expressed concern that the time commitment of full-time study meant that they were unable to work, “we worry sometime because we are studying full-time and not working yet”. Worrying about their livelihoods made it difficult for students to learn “[because we are] overthinking, we can’t concentrate on the things that we are going to do”. Another participant said, “here we feel like [attending English language classes] is compulsory even if we don’t have enough money”. In addition to worry, staying up late because of study affected some participants’ sleep, and therefore their health.

Nevertheless, participants valued the opportunity to learn to support future employment and their health. One participant said, “we need education… and information, what to eat, what not to eat, we need more information about healthy food.” Another added, “we don’t really know what is healthy eating… some information would be very helpful”. Participants also wanted information about topics such as exercise and sexual and reproductive health. One participant said it would be good to learn about “de facto or just boyfriend, and safe sex, those things can be included as well. You can get transmitted disease. Some people have HIV and don’t know.”

## 4. Discussion

Health remained a constant value for participants throughout their migration journey. This study highlights some of the changes refugees experienced upon migration to Australia and the positive and negative impact that these changes had on their health. Improved access to healthy food, settlement and livelihood support from the Australian government, access to healthcare, and strong social networks were amongst the strong influences on health for participants after arrival. These influences supported health-promoting behaviours that participants associated with the attainment of wellbeing. For instance, connections and linkages to their own ethnic communities played important roles in sharing information and knowledge, enacting health promoting behaviours such as physical activity and taking collective community action to address health issues.

Our findings are similar to studies of other refugee communities. For instance, in a study of a Bhutanese refugee community, Im and Rosenberg [41] discussed how social networks were an important setting for imparting health skills and knowledge, and not only resulted in individual behaviour change, but these changes had a flow-on effect to benefit the whole community. Like the Myanmar refugees in our study, the Bhutanese refugee community valued social connections and viewed this as inseparable from health and wellbeing [41]. In addition, social capital was built as community members helped and supported each other to continue engaging in and learning about health behaviour [41]. Our findings also echo the findings from Im’s study [42], as they illustrate how social connections are used as means of ‘collective coping’ in relation to health.

Participants in our study described the numerous difficulties and challenges during the settlement process, such as low literacy in the host language, challenges accessing services (including healthcare) and low income and unemployment. The challenges identified in this research echo that of other studies relating to refugee settlement [2,16,22,23,24,43]. Many of the challenges that refugees face stem from their transition to a new environment and the need to adapt to different systems, environment, people and more [44]. Therefore, using a strengths-based approach allowed for double-listening through exploration of health experiences while allowing space for trauma without re-traumatization [45,46]. In our study, we reported on refugees adjusting to the change in living environment and food supply, the need for further health information and the financial difficulties experienced. For instance, a key challenge to attaining health and wellbeing was the difficulty in obtaining employment due to limited English proficiency and lack of relevant qualifications and work experience. This in turn had an impact on participants’ mental health due to a conflict between studying English and working, which contributed to their financial difficulties. Whilst refugees receive settlement support upon arrival in Australia, it was not sufficient to address all of the challenges that refugees from Myanmar experienced in their daily lives.

Through sharing information and providing social and emotional support, our study highlights how members of the Myanmar community worked together to overcome health challenges. Our research builds on existing evidence that social support was a common strength and trait of refugee communities hailing from Asia [41,44]. This is an important area to recognise, as those who come from refugee backgrounds are often perceived to be ‘powerless’ and reliant on the provision of help and support from external service providers [47]. This perception likely stems from traditional and paternalistic models of care that view refugees as subordinates to professionals who are responsible for providing the services needed [47]. The issue with this perception is that it undermines the innate ability of all human beings to problem solve and discover their own solutions regardless of their social status in society. In fact, it has been argued that every community possesses strengths and assets that can be capitalised to initiate positive changes [48,49]. Through our research, we have identified numerous occasions where participants have created their own solutions to address challenges encountered whilst living life in Australia.

### 4.1. Implications for Practice and Considerations for Further Research

Our study looked at the inter-relationships between health, social connections, work and education. The complex nature of health and relationship with non-biomedical factors as highlighted in previous literature [11,12,13,21,22,23,24,25] was demonstrated in our research, and further supported the SEM, which challenges researchers and practitioners alike to think broadly about all influencers of health including social and environmental factors [29]. However, those currently implementing changes to improve the health and wellbeing of refugees are healthcare or social support service providers. We recognise the important work that these service providers are already doing to improve health and wellbeing outcomes of refugees. We also see the need for greater cross-sector collaboration to address broader social, economical and environmental influences of health for refugees. We call for greater support from policymakers to facilitate a more holistic approach to addressing the health and wellbeing needs for people from refugee backgrounds.

Whilst service provision is usually driven by service providers and their respective funding bodies, as well as policies behind them, our study gave examples of how refugees from Myanmar demonstrated understanding of service needs and gaps, and the roles that they played to problem solve when these gaps were present. For instance, some information and support needs were met by members of the Australian Myanmar community, by which community members took initiative to orientate new arrivals to living life in Australia, thus acting as a key resource for information. It is important for service providers to recognise community strengths as a resource and to consult closely with the community to ensure that service needs are appropriate and meet service needs and gaps. Service providers and policies should support refugee communities in driving and taking community-initiated action for their wellbeing.

### 4.2. Strengths and Limitations of this Study

Community members were integral in designing this research through their active involvement in the steering committee. Another significant strength was the trust that developed between the participants and the researchers, resulting in high participation rates in the focus groups (only three students declined). This trust was attributed to the rapport-building strategies, which were designed with community leaders, and involved a trusted individual introducing researchers to the students over a sustained period. Trauma-informed and strengths-based approaches also contributed to establishing trust while also affording the past and present health experiences of the participants to be explored without re-traumatization. Focus group participants represented a good cross section of ages (18 to 65 years), and the gender split was almost equal. A possible explanation of this is the nature of the setting where participants were recruited from (English language classes at Croydon SUT), highlighting that education is important across both genders as well as age groups.

We recognise that our study has some limitations. The main limitation is that focus groups were conducted in English and translated into several community languages through interpreters. To maximise data quality, qualified interpreters were employed and a researcher with experience conducting focus groups and working with interpreters was chosen as a moderator. In addition, the interpreters were briefed before each focus group to provide context, describe the purpose of the study and outline related focus group questions. In addition, data may have been influenced by differences in vocabulary between the two languages. For instance, there is no expression in Myanmar languages for ‘mental health’. We therefore tried to avoid such terms wherever possible, and used plain English throughout the focus groups. Due to the close-knit nature of the Myanmar community in Melbourne, participants may have withheld or modified information shared, if they had an existing relationship with the interpreters. Nonetheless, we believe that this was a minor limitation as participants discussed sensitive topics, such as betel nut chewing, in depth. The ethnic groups and religions recruited in the study reflected the Myanmar refugee student population living in the local catchment area. We recommend that other ethnic groups who have settled in other catchment areas such as the Kachin, Shan and Rohingya communities be included in future research. Although we consider high rates of participation a strength, it is possible that participants felt obliged to participate due to the power differential, despite our efforts to minimize this through the participatory nature of this study.

Our current study only examined the perspectives of health and wellbeing from Myanmar refugees. However, there are numerous service providers who presently provide health and wellbeing-related services to refugees in Melbourne, and the health needs of Myanmar refugees from the perspective of service providers remains unexplored in the literature. Further research exploring the latter is therefore warranted, offering opportunities to compare both perspectives in view of further improving healthcare services for all refugees.

## 5. Conclusions

This study outlined the inter-relationships between health, social connections, work and education from perspectives of refugees from Myanmar. The complex relationships between health and social, economic and environmental factors was in line with other existing literature. The study also outlines how people from Myanmar who are of a refugee background possess strengths that can be used to manage the various health challenges they face in their new environment. This approach is supported by assets-based or strengths-based community development literature and person-centred care approaches, while also being in line with trauma-informed care. Service providers in different sectors should work collaboratively to address the multitude of factors impacting health of refugees. Furthermore, service providers should recognise the strengths that people from refugee backgrounds possess and work closely with individuals and/or the community to support them in developing strategies and interventions tailored to their needs.

## Figures and Tables

**Table 1 ijerph-17-00121-t001:** Participant characteristics.

	Men (*n* = 12)	Women (*n* = 15)
	*n* (%)	*n* (%)
**Age**		
18–25	2 (16.7)	6 (40)
26–35	1 (8.3)	2 (13.3)
36–45	5 (41.7)	1 (6.7)
≥46	4 (33.3)	6 (46.7)
**No. of years lived in Australia**		
≤2 years	4 (33.3)	5 (33.3)
>2 years	8 (66.7)	10 (66.7)
**Martial status**		
Married	9 (75)	11 (73.3)
Not married	3 (25)	4 (26.7)
**No. of people living in household**		
1–2	1 (8.3)	0 (0)
3–4	2 (16.7)	5 (33.3)
5–6	6 (50)	3 (20)
>6	3 (25)	7 (46.7)

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
