# Peer review of "Living a Healthy Life in Australia: Exploring Influences on Health for Refugees from Myanmar"

_ijerph, 2019, doi:10.3390/ijerph17010121_

Round 1

Reviewer 1 Report

Thank you for an interesting paper and study looking at this important topic. I really enjoyed reading this paper, and your analysis of the key issues facing this cohort of people from Myanmar and refugee backgrounds.  I have some brief comments on the paper:

Abstract and Introduction: Your writing is clear and you provide a good introduction of the study. You have highlighted your rationale and key messages in the study. Literature review – This section of the paper is strong although brief, and your discussion of the literature and analysis of the key studies in this area here is good. Methods -This section is clear, and you have defined the rationale and provided depth for your decisions in relation to this project. I commend you on the co-production and collaborative work with the steering committee. Findings and Discussion: I enjoyed reading the findings and the quotes from the informants. You clearly identify and analyse the key themes and the implications. This is rich data and well presented. Conclusions: You make some good suggestions for implementation – for policy makers and service providers and link the key messages of your work back to the original aims. It looks like solid community development to me!

Great work and well done on this interesting study. I'm sure it will be of value to the community.

Author Response

Thank you for your comments.

Reviewer 2 Report

I have read with pleasure and great interest the article presented to the journal “Environmental Research and Public Health”  for its evaluation and possible publication.

It is an elegantly, well-organised, solidly structured and thoroughly researched study  in which the authors demonstrate a deep and detailed knowledge of the topic.

Before publication, I recommend to the authors to link a little better the theoretical framework to the analysis. In this way, a more fluid work would be achieved. Please, also, update the literature.

Author Response

Reviewer 2- I have read with pleasure and great interest the article presented to the journal “Environmental Research and Public Health”  for its evaluation and possible publication.

It is an elegantly, well-organised, solidly structured and thoroughly researched study  in which the authors demonstrate a deep and detailed knowledge of the topic.

Before publication, I recommend to the authors to link a little better the theoretical framework to the analysis. In this way, a more fluid work would be achieved. Please, also, update the literature

Response: Thank you for your comment. This was clarified in Section 2.4 Analysis, line 150-151. We have explained further how our results linked with the theoretical framework (lines 377-379). New up-to-date references have also been added (references 11-13, 18-19, 27).

Reviewer 3 Report

This study is well-formulated and -conducted, and the paper is generally well-written. I would like to make a few minor suggestions:

The authors focus on the assets and strengths of the refugees in the data analyses and interpretation. In the Methods and the Discussion sections, is it possible to elaborate how this may have shaped the analyses and results of this study? Line 64-- does a trauma-informed approach only focus on the strengths of the refugee participants? Line 345-357--  For example, is it possible that the difficulties of the refugee participants were overlooked due to the asset / strength based orientation of the study? Can the authors comment on how the experiences of the refugees were different to those of new immigrants who may not have experienced any trauma at all or may have experienced less trauma as compared to refugees?   Section 2.3 Data Collection-- is it possible to discuss here or in the Discussion section how power may have played a role in shaping the participant enrollment and generation of data? Specifically, as all participants were students and have presumably lower power status as compared to the steering committee and the research team, it seems possible that the power differential may have led to the high participation rate (due to trust or a sense of obligation?) and undermined/ biased the quality of the disclosure. Section 2.5 Rigour-- In member checking, did the process lead to confirmation of all themes? Was member checking done with the same student sample or an independent one, and what are the implications? Please clarify.   Are all quotations (e.g. line 199) formatted properly? Please double-check.

Further editing/ proofreading is suggested to make the paper read better: e.g. lines 265, 357-- please use professional editing if possible.

Author Response

Reviewer 3- This study is well-formulated and -conducted, and the paper is generally well-written. I would like to make a few minor suggestions:

The authors focus on the assets and strengths of the refugees in the data analyses and interpretation. In the Methods and the Discussion sections, is it possible to elaborate how this may have shaped the analyses and results of this study?

Response: Thank you for your comment. Both of these approaches are important strengths of this research, and this was acknowledged in Section 4.2. (line 402-404). We regard trauma-informed care approaches as an essential ethical consideration when working with people from refugee backgrounds.

Line 64-- does a trauma-informed approach only focus on the strengths of the refugee participants? Line 345-357--For example, is it possible that the difficulties of the refugee participants were overlooked due to the asset / strength based orientation of the study?

Response: Trauma-informed and strengths-based approach allow for double-listening of the strengths and the trauma in a safe space, i.e. without re-traumatization. This has been acknowledged in text with relevant references (line 350-352).

Can the authors comment on how the experiences of the refugees were different to those of new immigrants who may not have experienced any trauma at all or may have experienced less trauma as compared to refugees?

Response: Thank you for this comment. We appreciate that this would be an interesting area to contrast and explore further, however in the case of this research, our primary aim was to focus on highlighting current reported experiences of refugees from Myanmar. We used a trauma-informed lens to inform the study so that we as a research team would be sensitive in our approach, given the high chances that participants would have experienced some form of trauma related to their refugee experience. We do acknowledge though that it may be worthwhile comparing and contrasting experiences between refugees and immigrants in future research.

Section 2.3 Data Collection-- is it possible to discuss here or in the Discussion section how power may have played a role in shaping the participant enrolment and generation of data? Specifically, as all participants were students and have presumably lower power status as compared to the steering committee and the research team, it seems possible that the power differential may have led to the high participation rate (due to trust or a sense of obligation?) and undermined/ biased the quality of the disclosure.

Response: Thank you for your comment. This has been acknowledged as a potential limitation of this study (line 423-425).

Section 2.5 Rigour-- In member checking, did the process lead to confirmation of all themes? Was member checking done with the same student sample or an independent one, and what are the implications? Please clarify.  

Response: Thank you for your comment. This has been clarified in Section 2.5 (line 158-161).

Are all quotations (e.g. line 199) formatted properly? Please double-check. Further editing/ proofreading is suggested to make the paper read better: e.g. lines 265, 357-- please use professional editing if possible.

Response: Improvements have been made to formatting of quotations (Lines 204-206 and 215-217), as well as wording (Lines 277, 371-372).